# Molecular Characterization of Small Ruminant Lentiviruses Isolated from Polish Goats with Arthritis

**DOI:** 10.3390/v14040735

**Published:** 2022-03-31

**Authors:** Monika Olech, Anna Kycko, Jacek Kuźmak

**Affiliations:** 1Department of Swine Diseases, National Veterinary Research Institute, 24-100 Puławy, Poland; 2Department of Biochemistry, National Veterinary Research Institute, 24-100 Puławy, Poland; jkuzmak@piwet.pulawy.pl; 3Department of Pathology, National Veterinary Research Institute, 24-100 Puławy, Poland; anna.kycko@piwet.pulawy.pl

**Keywords:** goat, small ruminant lentivirus, SRLV, proviral load, arthritis, pathogenicity, histopathology, genetic characterization

## Abstract

Previous studies revealed that the small ruminant lentivirus (SRLV) population in Poland is highly heterogeneous. All SRLVs detected from Polish sheep and goats so far have belonged to subtypes B1, B2, A1, A5, A12, A13, A16, A17, A18, A23 and A24. However, all characterized strains originated from asymptomatic animals. This is the first study that characterizes the molecular properties of SRLVs isolated from different organs of six arthritic goats. Segments from three genomic regions (*gag*, LTR and *env*) were analyzed. In addition, we quantified the SRLV proviral load in the blood and different organs and examined its association with different degrees of histopathological lesions. All sequences obtained from the goats involved in this study were homogeneous, showing an average degree of variability of 4.8%, 3.7% and 8.8% for *gag*, LTR and *env*, respectively. Phylogenetic analysis revealed that the sequences from the analyzed goats were clustered within SRLVs group A and formed a new subtype within this group, tentatively named A27. The histopathological examination of the lung, mammary gland, synovial membranes of joints and brain of the analyzed goats revealed evidence of inflammatory processes associated with SRLV infection, which was confirmed by positive immunohistochemistry assays. No significant correlation was observed between histological features and alterations in the sequences from different tissues. No tissue-specific signature pattern was identified. It was shown that animals with a higher proviral load showed more lesion severity in various SRLV-affected tissues, indicating a positive association between these two parameters. Our results also revealed differences in the SRLV load between animals even though the sequences derived from all of the goats were closely related, suggesting that host factors may restrict and control viral replication. This study provides new information about SRLV variants isolated from arthritic goats; however, more studies, including the isolation and characterization of biological properties of these viruses, should be performed to evaluate their pathogenic potential.

## 1. Introduction

Maedi visna virus (MVV) and Caprine arthritis encephalitis virus (CAEV) are two single-stranded RNA viruses belonging to the Lentivirus genus of the *Retroviridae* family that cause a lifelong persistent infection in sheep and goats. Initially, MVV and CAEV were considered pathogens specific to sheep and goats, respectively. Recently, due to their ability to cross the species barrier and their genomic and structural similarities, these two viruses are considered a single group called small ruminant lentiviruses (SRLVs). Transmission occurs mainly via the respiratory route and through the ingestion of infected colostrum/milk [1]. The main target cells of the SRLVs are monocytes/macrophages and dendritic cells, and viral replication is restricted until the maturation of monocytes to macrophages [2,3]. SRLVs cause progressive inflammatory syndromes in different host organs, including the lungs, udders, joints and central nervous system. On infection, animals may remain asymptomatic carriers for life, and only a small proportion of infected animals develop clinical signs such as interstitial pneumonia, indurative mastitis (“hard udder”), arthritis, dyspnea and, more rarely, encephalitis, ataxia or paralysis. In goats, arthritis, indurative mastitis and, more rarely, encephalomyelitis in young kids occur. Occasionally, the infected ones may develop subclinical interstitial pneumonia and progressive dyspnea [1]. Both asymptomatic and symptomatic animals can transmit the virus. Disease progression is usually slow, and the clinical course and outcome of infection are associated with the genetic background of the animals and the virulence of the infecting strain [4,5]. Histopathologically, the different SRLV disease forms involve lesions with infiltration of mononuclear cells (macrophages and lymphocytes), resulting in inflammation, fibrosis and thickening of the tissues. The severity of the histological lesions is shown to be more intense in tissues with higher viral and proviral loads [6,7]; however, the mechanism that regulates viral replication and leads to increased viral load and more severe disease outcomes is still unclear.

SRLVs are genetically heterogeneous, which may account, in part, for the different patterns of clinical diseases. SRLVs are currently divided into five phylogenetic groups, A–E, and further divided into subtypes [8,9]. The mechanism contributing to the distinct tissue specificity of SRLVs remains poorly understood. However, two areas that may contribute to tropism include long terminal repeats (LTRs), particularly the U3 promoter region, and the C-terminus of the Env protein.

The SRLV genome, which is integrated into host cells in the form of a provirus, contains three structural genes (*gag*, *pol* and *env*) and three regulatory genes (*vpr-like*, *rev* and *vif*) flanked by non-coding long terminal repeat (LTR) regions. The *gag* gene encodes the capsid protein, which is very conserved and contains linear epitopes, and for this reason, it is used for phylogeny and serological diagnosis. The LTR is divided into three regions, U3, R, and U5, and the U3 region contains a promoter/enhancer sequence and transcription factor binding sites, such as activator proteins (AP)-1, AP-4, AML (vis), γ-interferon activated site (GAS) and TNF-activated site (TAS) [10,11]. Several mutations, deletions or duplications in the LTR of SRLVs were shown to be associated with virulence, cellular tropism and pathogenesis [12,13,14,15,16]. Alternatively, envelope glycoproteins may also play a role in cellular host range, infectivity and disease progression. It was shown that hypervariable regions of the *env* gene involved in compartmentalization are implicated in tropism. In the SRLV genome, five variable (V1–V5) regions were previously identified [17]. The presence of different SRLV sequences in the V4 region in different organs suggests that SRLVs can undergo variations in this region during early infection, giving rise to different viral subpopulations. A study on the compartmentalization of SRLVs revealed the presence of different sequences in the blood, lungs, mammary glands, central nervous system and colostrum [18,19,20].

Previous studies revealed that the population of SRLVs in Poland is highly heterogeneous. The SRLVs detected so far from polish sheep and goats belonged to subtypes B1, B2, A1, A5, A12, A13, A16, A17, A18, A23 and A24 [8]. However, all characterized strains originated from asymptomatic animals. In this study, we describe SRLV sequences isolated from different organs of arthritic goats for the first time. We determined partial *gag*, *env* and LTR sequences, investigated their phylogenetic relationships with sequences of other known SRLV strains, quantified the SRLVs’ proviral load in the blood and different organs and examined its association with different degrees of histopathological lesions.

## 2. Materials and Methods

### 2.1. Animals, Serology and Tissue Sample Collection

Six naturally SRLV-infected adult goats originating from northern Poland were used for sample collection. The animals belonged to a flock with a high seroprevalence of SRLV infection as were assessed through the use of commercial test ID Screen MVV/CAEV Indirect Screening (IDvet, Grabels, France). The goats presented as thin bodied (enlargement of the carpal joint, leg weakness, ataxia) and were humanely euthanized at the owner’s request due to deterioration in their quality of life. Prior to euthanasia, peripheral blood mononuclear cells (PBMCs) were isolated from EDTA-anticoagulated blood on a Ficoll-Hypaque gradient. Blood collection was approved (no. 37/2016) by the Local Ethical Committee on Animal Testing at the University of Life Sciences in Lublin (Poland). A complete set of tissues, including joints, lungs, mammary glands, brain and choroid plexus, were collected post-mortem in a thermos flask (+4 °C) in individual containers filled with Dulbecco’s phosphate-buffered saline (DPBS) for viral isolation. Additionally, samples from the brain, including choroid plexus, lungs and mammary glands, were collected for nucleic acid extraction and histopathological analysis. In addition, synovial fluids and synovial membranes obtained from joints were used for virus isolation and molecular characterization, respectively.

### 2.2. Histopathology and Immunohistochemistry

Tissue samples were promptly fixed in 10% buffered formalin, then routinely processed, embedded in paraffin and cut on a microtome for histopathology and immunohistochemistry (IHC) examination. For histopathology, 4 μm thick slides were stained with hematoxylin and eosin (HE). The intensity of the lesions in the target organs was estimated independently by two pathologists. The severity of arthritis/synovitis among the tissues from the examined goats was scored according to the grading system proposed by Cheevers et al. [21]. The severity of the inflammatory infiltration in the examined sections of the lungs and mammary glands was scored as follows: 0—No changes, 1—Mild diffuse inflammatory infiltration, 2—Moderate diffuse inflammatory infiltration, 3—Diffuse inflammatory infiltration with the formation of lymphoid follicles. The severity of lesions observed in the brain was scored as follows: 0—No changes, 1—Mild lymphoplasmacytic infiltration within choroid plexus, 2—Moderate lymphoplasmacytic infiltration within choroid plexus and focal perivascular cuffings, mild gliosis.

Select tissues with lesions histologically consistent with SRLV-associated inflammation were further evaluated by an immunohistochemistry assay to detect the SRLV p28 Gag antigen. IHC was performed using the DAB/HRP Envision system (DAKO, Glostrup, Denmark). Briefly, 4 μm thick paraffin-embedded tissue sections were cut and mounted on positively charged slides. After overnight air-drying, the slides were deparaffinized, rehydrated using xylene and graded alcohols and then treated with a 3% hydrogen peroxide solution for 10 min to block the activity of endogenous peroxidases. Antigen retrieval was carried out by treatment in a pressure cooker (110 °C) with citrate buffer pH 6 for 20 min. The slides were then incubated with the CAEV10A1 primary antibody (VMRD) diluted 1:100 for 60 min at room temperature (RT), followed by incubation with a peroxidase-conjugated polymer as a secondary antibody (Dako REAL EnVision Detection System, K5007, DAKO, Glostrup, Denmark) for 30 min. After color development with diaminobenzidine solution, sections were counterstained with Mayer’s hematoxylin, dehydrated and mounted. Sections incubated with PBS instead of the primary antibody were used to confirm the specificity of the staining. The tissues were analyzed under a light microscope (Axiolab, Zeiss, Germany) for the presence of positive labeling.

### 2.3. DNA Extraction, Amplification and Sequencing

Genomic DNA was extracted from tissue samples using a NucleoSpin Blood Quick Pure Kit (Macherey-Nagel GmbH & Co KG, Dueren, Germany), according to the manufacturer’s instructions. The quality and quantity of the DNA were checked in a Nanophotometer (Implen GmbH, Munich, Germany). The CA (625 bp) fragment of the *gag* gene fragment, the V4V5 fragment of the *env* gene and the U3-R fragment of the LTR region were amplified by nested PCRs, as previously described [8]. A water template negative control was run parallel with each PCR reaction set. PCR products were purified using NucleoSpin Gel and PCR Clean-up (Marcherey-Nagel, GmbH 7 Co., Hamburg, Germany) and directly sequenced on a 3730 xl DNA Analyzer (Applied Biosystems, Foster City, CA, USA) using a BigDye Terminator v3.1 Cycle Sequencing kit. The obtained SRLV sequences were edited and aligned against known reference sequences using the MUSCLE multiple-alignment software provided as part of Geneious Pro 5.3 (Biomatters Ltd., Auckland, New Zealand). All novel sequences reported in this study were submitted to the Gen-Bank database under accession numbers: OM517045-OM517079 for the *gag* sequences, OM517080-OM517094 for the *env* sequences and OM517095-OM517127 for the LTR. The best-fitting nucleotide substitution models were estimated, and the GTR, TN93 and K2 with the gamma distribution rates were used to infer a phylogenetic tree using the maximum likelihood (ML) method for the *env*, *gag* and LTR sequences, respectively. The reliability of the phylogenetic relationships was evaluated by nonparametric bootstrap analysis with 1000 iterations. Alignment, model testing, and tree building were performed using the MEGA 6 application [22]. Nucleotide and amino acid sequence percent identity (percentage of bases/residues which are identical) was estimated using the Geneious software, while pairwise genetic distances were calculated with the MEGA 6 software.

### 2.4. Proviral Load Quantification

Proviral DNA was quantified by real-time PCR using Rotor-Gene Q Series ver. 2.0.3 (Qiagen, Hilden, Germany) with primers and a probe specifically designed based on the *gag* sequences obtained for the SRLV subtype described in this study. The sequence of the forward and reverse primers and probe were CAPBWF (5′GAAGCAGAAAGGTGGGTAAGA 3′), CAPBWR (5′ TTGCGATGCCTGTTGATTTG3′) and CAPBWP (5′ 6-FAM-ATCCACTGTGAGGACATTTGGCCC -BHQ-1 3′), respectively. The 50 µL reaction mixture contained 25 μL of 2× QuantiTect Multiplex NoROX PCR buffer (Qiagen, Hilden, Germany), 400 nM of each primer, 200 nM of the TaqMan probe and 500 ng of extracted genomic DNA. The cycling protocol involved an initial incubation for polymerase activation at 95 °C for 15 min, followed by 45 cycles of denaturation at 94 °C for 60 s and annealing/extension at 60 °C for 60 s. A non-template control (DEPC H2O) was included in each run. Standard curves were generated using a 10-fold serial dilution from 10^8^ to 10 copies of the reference plasmid encompassing the target *gag* region. All samples were tested in duplicate, and the results were expressed as a mean copy number of provirus per 500 ng of genomic DNA of each goat.

## 3. Results

### 3.1. Histopathology and Immunohistochemistry (IHC)

All of the goats presented macroscopic lesions compatible with chronic proliferative arthritis that consisted of enlarged carpal joints. Histopathological changes observed in the joints involved synoviocyte proliferation and synovial membrane thickening with the formation of papillary Villar projections, accompanied by lymphoplasmacytic infiltrations of the subsynovial connective tissue and perivascular spaces (Figure 1E) as well as diffuse fibrosis. Necrosis with focal calcification was occasionally observed within the fibrous capsule. The severity of the lesions observed in the joints is summarized in Table 1.

In the lungs, mild changes characteristic of interstitial pneumonia, including interstitial lymphocytic infiltrations and thickening of the alveolar septa, were observed (Figure 1C). In the mammary glands, around the acini and the lactiferous ducts, diffuse mononuclear inflammatory infiltrates of variable intensity were present, accompanied by multifocal inspissation and the occasional mineralization of the duct contents (Figure 1G). Finally, lesions in the brain were scant and seen only in three out of six analyzed goats. They were characterized by mild lymphoplasmacytic infiltration of the choroid plexus and, rarely, focal perivascular cuffings and gliosis in the brainstem (Figure 1A,B). The scored severity of the inflammatory infiltration in the examined sections of the lungs, mammary glands and brains of the goats is shown in Table 1.

Immunohistochemical examination showed positive staining for SRLV p28 in the articular tissues in several inflammatory cells, in the lungs in several bronchiolar epithelial cells and single mononuclear cells, as well as in single macrophage-like cells in the mammary glands (Figure 1). IHC performed on the brain tissues was negative. There was no positive immunolabeling detected in any of the examined brain sections.

### 3.2. Phylogenetic Analysis

Total DNA isolated from the peripheral blood mononuclear cell (PBMC), mammary gland, lung, synovial membrane and brain of all goats were used as templates for the PCR amplification of the *gag* gene encoding the capsid protein. The band corresponding to the expected 625 bp PCR product was detected in all tissue samples of the animals except the lung-derived samples originating from goat 5. The obtained sequences were aligned with other published sequences representing SRLV genotypes described to date and subjected to phylogenetic analysis. The maximum-likelihood (ML) tree revealed that all analyzed sequences were closely related to each other (mean genetic distance of 4.8%) and that the sequences derived from different tissues of the same animal clustered together. The mean nucleotide distance of sequences derived from different tissues of each goat did not exceed 1.3% (ranging from 0.2% to 1.3%). Interestingly, all of the analyzed sequences formed a new cluster within group A, which could be tentatively named A27. The affiliation of the new cluster, A27, was supported with high bootstrap values ≥85 (Figure 2). The mean genetic distances between sequences forming the new subtype and sequences representing other subtypes representative of genotype A varied from 10.0% to 21.7% (Table 2). Polish SRLV A27 nucleotide sequences showed the highest nucleotide sequence identity, with strains representative of subtype A13. To evaluate the robustness of our analysis, we also performed a phylogenetic analysis using the neighbor-joining method (Appendix A), which resulted in the same classification, supporting the existence of the new subtype.

In order to confirm the genetic assessment of the sequences obtained from the analyzed goats, we amplified and sequenced a genomic fragment encompassing a highly variable region of the envelope gene (*env*) encoding for the carboxy-terminus portion its surface subunit (SU). Unfortunately, amplification of this fragment succeeded only for the lung-derived samples of goat 4 and 6, brain-derived samples of goat 4 and 5, choroid plexus-derived samples of goat 5, mammary gland-derived samples of goat 4, 5 and 6, synovial membrane-derived samples of goat 2, 4, 5, 6 and 7 and PBMC-derived samples of goat 2 and 5. The phylogenetic tree (Figure 3) confirmed that the sequences originating from the goats analyzed in this study belonged to the new subtype A27. The mean nucleotide distance between sequences belonging to the A27 and other subtypes within group A ranged from 23.7% to 31.8%. The mean genetic distance of the *env* sequences originating from all goats was 8.8% and ranged from 4.7% to 12.6%. Within each animal, the mean nucleotide distance of sequences from different tissues did not exceed 7.5% (ranging from 0.8% to 7.5%).

### 3.3. Comparative Analysis of Immunodominant Regions

Nucleotide sequences of the *gag* and *env* genes were translated into amino acid (aa) sequences and aligned with the aa sequences of reference strains: Cork and K1514. The pairwise percent identity of the *gag* amino acid sequences of the SRLVs obtained from the tested goats was high and ranged from 91.7% to 100%. The pairwise percent identity of sequences originating from the same animal ranged from 97.4% to 100%. Furthermore, the analyzed sequences shared 84.2–86.8% and 93.1–96.1% amino acid sequence identity with the Cork and K1514 strains, respectively. Analysis revealed that all CA sequences originating from all tested goats had the asparagine-valine (NV) motif, which is specific to MVV-like (group A) sequences (Figure 4). Epitope 2 was fully conserved in all sequences originating from different tissues of all goats. Sequences of epitope 3 were also quite conserved. Only glutamic acid (E) replaced lysine (K) in sequences derived from the choroid plexus and brain of goat 6, and arginine (R) replaced lysine (K) in all sequences derived from goat 4. More alterations were found in the major homology region (MHR), which is highly conserved in all retroviruses. No tissue-specific signature pattern was identified in the *gag* sequences. Most of the substitutions were specific for each animal. In particular, all sequences from goat 3 had the substitution of lysine (K) instead of arginine (R) at position 137 compared to the sequences originating from the other goats, aspartic acid (D) instead of glutamic acid (E) and histidine (H) instead of glutamine (Q) in positions 7 and 98, respectively. All sequences originating from different tissues of goat 2 had the unique substitutions ^S^41^N^, ^K^129^R^ and ^K^139^R^, while all sequences from goat 4 had the unique substitutions ^K^41^R^, ^E^46^D^, ^A^101^S^, ^N^161^T^, ^A^177^S^ and ^S^178^T^. The sequences from goat 7 had the unique substitutions ^M^118^T^, ^N^141^S^, ^V^188^T^ and ^T^191^N^, while the sequences from goats 5 and 6 had the unique substitution ^I^148^V^. Furthermore, all sequences from goats 2 and 3 had the specific substitution isoleucine (I) instead of threonine (T) at position 10 compared to the sequences from the other goats.

Next, the deduced amino acid sequences of the immunodominant regions of the surface glycoproteins (SU) were aligned with the corresponding reference sequences of parental strains Cork and K1514 (Figure 5). The *env* aa sequences obtained from the goats were more heterogeneous, showing 77.6–98.3% similarity to each other and 65.8–71.0% and 77.0–79.0% to the Cork and K1514 strains, respectively. Most of the variation was observed in the variable region (V4), especially in the highly variable region (HV2) (Figure 5). Sequences of epitope SU5 were more conserved among the sequences derived from the analyzed goats and differed prominently from the SU5 sequences of the Cork and K1514 strains and other strains representing known subtypes (Appendix A). Only the sequence originating from the choroid plexus of goat 5 differed from the sequences originating from the other goats analyzed in this study. All sequences had a perfectly conserved motif located at the N-terminus part (VRAYTYGV) and more the variable motif at the C-part.

### 3.4. Analysis of LTR Sequences

LTR sequences from the lungs, choroid plexus, mammary glands, synovial membrane and PBMC of all six goats and from the brain of goats 4, 5 and 7 were successfully amplified and were aligned with the corresponding sequences of SRLV strains with defined virulence, including sequences of the virulent B1 strain Cork [24], A2/A3 strain 697 isolated from sheep with neurological symptoms [25], neurovirulent A1 strain Kv1772 [13], B2 strain 496 isolated from arthritic sheep [26] as well as to the attenuated A4 strain 6221 [27].

The LTR sequences originating from all goats were closely related to each other. The mean genetic distance of the LTR sequences was 3.9%. Only the sequence derived from the choroid plexus of goat 4 was quite divergent from the rest of the sequences. The genetic distance of this sequence, compared to the other sequence from different tissues of all analyzed goats, ranged from 14.5% to 16.8%. The mean nucleotide distance of the sequences from different tissues of each goat did not exceed 4.3%. Only in the case of goat 4 the mean nucleotide distance of sequences from different tissues was 6.5% and ranged from 0.5% to 16%. Phylogenetic analysis revealed that sequences derived from different tissues of the same animal generally clustered together (Figure 6). Only goat 4 had distinct sequences isolated from the choroid plexus and synovial membrane; goat 5 had a distinct sequence isolated from the lungs, while sequences isolated from the PBMC and mammary gland of goat 6 were quite different from the sequence isolated from the choroid plexus, synovial membrane and lungs.

Multiple conserved motifs were identified within the LTR sequences, including AP-1 sites, AML, AP-4, TATA box, poly A, CAAAT, GAS and TAS. Sequences corresponding to the TATA box, AP-4, GAS and polyadenylation signal (poly A) were quite conserved among all sequences (Figure 7). Almost all sequences originating from the analyzed goats had a unique T to A substitution in the fifth position of the TATA box. Five different sites of AP-1 (TGACAC, G/TAGTCA, TCATGTA, TTAG/AGTCA, ATA/GAT/CTA/GT/G) were observed in sequences derived from all goats analyzed in this study. Furthermore, all tissue sequences from all goats presented two highly conserved AML(vis) sites. Some point mutations were observed in the TAS region; however, the sequences originating from different tissues of the same animal were very homologous. Two deletions in the U3 region of the A5 strains were noted; one (10 nt) located near the 5′ end of the U3 region, and the second (21 nt) located in the central region of the U3. The presence of these deletions was also shown in attenuated strain 6221 as well as in virulent strain 697. The sequence derived from the choroid plexus of goat 4 had a 12-nt deletion in the R region, while all sequences from goat 3 had a 9-nt deletion in this track. Similar deletions were present in virulent strain 697 as well as attenuated strain 6221. The number of repeats of the CAAAT sequence was different depending on the strains. Strain Kv1772 had three copies of the CAAAT sequence, while Cork, strain 496 (B2) and 497 (A2/A3) had only one copy of the CAAAT, and strain 6221, representing subtype A4, had no copy. The number of CAAAT sequences derived from different tissues of the analyzed goats varied from 1 to 3. Only the sequences derived from the mammary gland and brain of goat 5 had three copies of CAAAT.

### 3.5. Provirus Detection and Quantification

The proviral load in the blood and different organs of infected animals was quantified by real-time PCR using primers and probes specifically designed based on the *gag* gene sequences of SRLVs described in this study. The number of provirus copies was determined by comparing the Ct values obtained by amplifying *gag* fragment in a particular sample to the standard curve generated after amplification of scale-down dilutions of plasmid DNA containing the targeted region. A new standard curve was generated for every experiment, showing a mean efficiency value close to 100% and a mean R-value of R = 0.998. The provirus copy numbers differed between the animals and the tissue analyzed (Figure 8). The highest proviral load was detected in the synovial membrane of goat 7 (4618 copies per 500 ng of genomic DNA). In goats 2, 3, 4 and 7, the highest proviral load was found in the synovial membrane cells. The copy number detected in these samples ranged from 47 to 3466. In goats 6 and 5, the highest proviral load was found in the lung and choroid plexus, respectively. Brain samples were positive only in four out of six goats and showed very low copy numbers (>4 copies per 500 ng of genomic DNA). The proviral load (>8 copies per 500 ng of genomic DNA) detected in the choroid plexus also was very low in 4 out of 6 analyzed goats, while in goats 5 and 7, the number of SRLV provirus copies detected in this tissue was 2233 and 65, respectively. In the mammary gland, PBMC and lung samples, the SRLV copy number varied from 44 to 351, from 9 to 1158 and from 9 to 820, respectively.

## 4. Discussion

In this study, we revealed, for the first time, the *gag*, LTR and *env* sequences of SRLVs detected in Polish goats with clinical signs of arthritis and determined their phylogenetic relationship in the context of known SRLV sequences. In addition, we examined the potential association between the degree of histopathological changes and provirus concentration in different organs of the infected goats.

All sequences obtained from the goats involved in this study were homogeneous, showing an average degree of variability of 4.8%, 3.7% and 8.8% for the *gag*, LTR and *env* sequences, respectively. Phylogenetic analysis on the basis of the *gag* and *env* sequences revealed that the sequences from the analyzed goats clustered within SRLV group A and formed a new subtype within this group, tentatively named A27. The SRLVs isolated until now from asymptomatic sheep and goats in Poland belonged to the well-known subtypes B1, B2, A1, A5 and A16, as well as subtypes A12, A13, A17, A18, A23 and A24, detected only in Poland [8,28,29,30,31]. The sequences obtained in this study were grouped in a cluster with a significant divergence from the other strains isolated from goats and sheep from Poland. The sequences were closely related to sequences belonging to subtype A13, which was previously detected only in sheep from Poland, suggesting their common origin. It was not known if the examined goats had contact with sheep, but infection with A subtypes in goats was reported [9,27], also in Poland, where the circulation of subtypes A1, A5, A12, A16, A17 and A23 [8,28,29,31] was detected in goats.

Indirect serological assays usually use the capsid protein as the antigen, so analysis of these sequences is useful for detecting antigenic variability, which may affect the sensitivity of serological tests. The primers used in this study allowed the sequencing of two major immunodominant epitopes identified by Rosati et al. [32] and the major homology region (MHR). Immunodominant epitope 2 was fully conserved in all sequences originating from different tissues of all goats, and only minor variations were found in sequences of epitope 3. This high conservation is important for the maintenance of cross-reactivity in serological tests based on Gag-derived antigen [17,33]. More alterations were observed in the major homology region (MHR), suggesting that this region is quite variable. Therefore, our results are in line with other studies demonstrating the variability of the MHR region [8,34]. No tissue-specific signature pattern was identified in the *gag* sequences. Most of the substitutions were specific to each animal, suggesting that these changes could have arisen as a result of long-term host–virus adaptation and evolution.

The *env* aa sequences obtained from the analyzed goats were more heterogeneous. Most amino acid substitution was observed in variable region V4, especially in highly variable region HV2, previously proposed to be part of a variable, conformational neutralization epitope [35,36]. It is proposed that the V4 region forms a highly constrained and surface-exposed domain and that a cysteine loop may have an analogous function to the V3 domain of HIV-1. Recently, a ‘signature pattern’ related to the SRLV genotype and different clinical statuses in sheep and goats was found in the V4 region. Possible signature patterns were detected at positions 54, 78, 79 and 80 in the sequence loop of the V4 region. However, the most important statistical significance was found at position 54, where residue N occurred only in sequences derived from arthritic animals infected with genotype B, while residue T and G occurred only in sequences derived from asymptomatic animals infected with genotype A and B, respectively [37]. Our results showed the presence of residue T at position 54 in all sequences derived from the different tissues of the analyzed goats. This may suggest that this position is not associated with clinical signs but may be associated with SRLV genotypes as all sequences with residue T at position 54 are derived only from animals infected with genotype A. We did not find a tissue-specific signature pattern in the *env* sequences. However, we were unable to obtain all sequences derived from the different tissues of the analyzed goats. PCR amplification detected *env* proviral DNA mostly in synovial membrane tissues, whereas samples from other tissues yielded limited positive results. This indicates a high variability of *env* sequences in different tissues, which may arise from compartmentalization and rapid sequence evolution during SRLV infection [18,19]. A study on SRLV compartmentalization revealed the presence of different sequences in the blood, lungs, mammary glands, central nervous system and colostrum [18,19,20]. Our study revealed limited genetic variability within the host. Only the sequence originating from the choroid plexus of goat 5 differed from the sequences originating from the other tissues of this goat as well as sequences originating from other goats. Moreover, the fact that the highest SRLV proviral load and moderate lesions were only observed in the choroid plexus of this goat may suggest that *env* sequences may be involved in tissue tropism, viral compartmentalization and pathology.

The U3 region of lentiviruses contains promoter sequences and transcription factor binding sites important for virus transcription and replication. It was demonstrated that variability in the LTR sequence is associated with tissue tropism and disease outcomes in a number of retroviruses [13,38,39]. However, in this study, we did not identify any tissue-specific motifs. Sequences isolated from the same tissue in different goats did not aggregate into a phylogenetic cluster. Our results revealed a rather coherent grouping of sequences obtained from different anatomical compartments of the same animals, indicating that the LTR sequences are not good predictors of viral tissue tropism. Sequences corresponding to the TATA box, AP-4, AML (vis) and polyadenylation signal (poly A) were conserved among all sequences derived from different tissues of the analyzed goats. Furthermore, our results revealed that almost all sequences derived from different tissues of the analyzed goats had a unique T to A substitution in the fifth position of the TATA box. The same mutation was also noted for Polish strains belonging to subtype A17 [8], but the meaning of this mutation is unknown. More alterations were observed in sequences of AP-1 sites. Additionally, GAS and TAS response elements for some cytokines were identified in the U3 region of all sequences. It was demonstrated that the pro-inflammatory cytokines, IFN-γ and tumor necrosis factor (TNF)-α, which are increased in goats with arthritis, activate viral transcription through the GAS and TAS sites [40]. In our study, GAS was identical in all sequences derived from the goats. These findings are compatible with previous studies showing absolute conservation of the GAS region in 40 of 41 viral promoter sequences, suggesting that this motif is important for viral pathogenesis [16]. The TAS region was less conserved; however, sequences originating from different tissues of the same animal were very homologous. Generally, our results are consistent with previous studies demonstrating only minimal variation in U3 sequences derived from different anatomical sites of a single host and that the TAS and AP-1 sites are the least conserved transcriptional features of the U3 region [10,15,16,41].

Previous studies also showed that duplication in the LTR U3 region of Icelandic neurological strains is a determinant of cell tropism. It was demonstrated that deletion in the duplicated sequence in this region caused deficient virus replication in choroid plexus cells, indicating that this duplication may be associated with neurovirulence [13]. Our results revealed the absence of 10-nt and 21-nt insertions in the U3 region of all sequences originating from the goats analyzed in this study, insertions that were present only in the Kv1772 strain. The lack of a 21-nt repeat resulted in a reduction of potential transcription factor binding sites (AP-1 and AML(vis)) and the CAAAT sequence located in this repeat. No putative transcription factor binding domains were identified within the 10-nt deletion. However, the presence of these two deletions was also shown in attenuated strain 6221 as well as in neuropathogenic strain 697, suggesting that these insertions are not related to a particular host tropism. Furthermore, our results revealed that the sequence derived from the choroid plexus of goat 4 had a 12-nt deletion in the R region, while all sequences from different tissues of goat 3 had a 9-nt deletion in this track. This deletion was previously described by Angelopoulos et al. [41], who suggested that the presence of this specific deletion may be associated with lower pathogenicity of SRLV strains. However, no substantial difference was found in the LTR region of small ruminants with and without clinical signs of disease. Our results revealed that a similar deletion was present in sequences of attenuated strain 6221 and strain 697 isolated from sheep from a neurological outbreak. Moreover, this deletion was found previously in the R region of the Polish strains isolated from clinically healthy sheep and goats [8] and clinically affected sheep from Spain [26], suggesting that factors besides this deletion are involved in avoiding the appearance of clinical symptoms. Generally, all of these observations are in line with the observations of Blatti-Cardinaux et al., who indicated that the role of LTR in tissue tropism is still unclear, suggesting that other viral sequences might be involved in tissue tropism and pathology [42].

SRLV proviral DNA was detected in all examined goats confirming the persistent nature of SRLV infection. In four out of six goats, the highest proviral load was found in the synovial membrane cells, confirming the tropism of isolated viruses to the joints. The relatively high proviral load was also observed in the mammary glands of all goats and the lungs of three goats. An increased proviral load in these affected organs was expected since they are the primary target organs of SRLVs and are directly involved in the efficient transmission of SRLVs through the colostrum, milk and respiratory exudates [43,44]. Several types of cells in the mammary glands, including macrophages and epithelial cells, harbor the virus playing an important role in its transmission to offspring. Additionally, SRLVs belonging to genotypes A4 and E1, which express low pathogenicity for goats, were abundantly presented in the mammary glands, confirming that lactogenic transmission is the main route of infection of SRLVs in this animal species [43,44]. It was also shown that SRLV subtype A is more efficiently transmitted to offspring, suggesting that A group-derived subtypes are particularly efficient in lactogenic transmission [45]. The lowest proviral load was detected in DNA extracted from the choroid plexus and brain cells, suggesting poor neurotropism of these viruses. The highest proviral load was only found in the choroid plexus cells of goat 5; however, no neurological symptoms were observed. High proviral loads in the absence of overt clinical signs in the infected animals are similar to the situation observed in simian immunodeficiency virus (SIV)-infected natural hosts and are probably related to the long coevolution between this lentivirus and their respective hosts [43,46]. Studies on neurovirulent variants of SIV showed that macrophage tropism is not sufficient for neurological disease [46].

Many studies describe a phylogenetic analysis of SRLVs, but only a few of them describe the histopathological lesions observed in the target organs of the infected goats/sheep [6,34,41,43,47]. The pathogenesis of the SRLV strains found in arthritic goats in Poland was not characterized in any case. The histopathological examination of the lungs, mammary glands, synovial membranes of joints and brain of the analyzed goats revealed evidence of inflammatory processes associated with SRLV infection, which was confirmed by positive immunohistochemistry assays, amplification and sequencing. It was shown that animals with a higher proviral load showed more lesion severity in various SRLV-affected tissues, indicating that the proviral load is positively correlated with the presence and severity of clinical disease symptoms [6,7]. Therefore, the goal of this work was to examine SRLVs’ target organs for their proviral load and the potential presence of histopathological lesions. Comparative analysis of the distribution of lesions according to the lesion degree and proviral load performed in this study indicated a positive association between these two parameters. This was most clearly seen in the case of the synovial membrane and brain tissues. The more severe lesions were observed in the synovial membrane of three goats, which indeed showed very high proviral loads. Furthermore, moderate lymphoplasmacytic infiltration within the choroid plexus and focal perivascular cuffings with mild gliosis was only observed in two goats, which showed higher proviral loads. Our results also revealed differences in the SRLV loads between animals even though the sequences derived from all of the goats were closely related. Goats 3, 4 and 6 had relatively lower proviral loads in all tissue samples, while goats 2, 5 and 7 had higher overall proviral loads. This different reactivity to SRLV infection suggests that host factors may restrict and control viral replication. We may speculate that goats with a lower proviral load mounted an immune response that was more efficient, which resulted in a reduced viral load. Several works identified immune response loci that could influence resistance/susceptibility to SRLV infection and disease, providing evidence that genetic factors might modulate the outcome [48,49,50,51,52,53,54,55,56]. It was demonstrated that host control of infection with SRLVs, including the provirus level, may have a genetic basis. Molecular studies in humans infected with HIV pointed to the role of SNPs in the Toll-like receptor (TLR) gene, especially in TLR7, and their association with viral load. In infection with SRLVs, it was also demonstrated that some polymorphisms were significantly associated with the SRLV provirus concentration. The obtained results suggest that SNPs of TLR7/8 genes may induce differential innate immune responses toward SRLVs, affecting the proviral concentration and, thereby, disease progression [57].

In conclusion, this is the first study investigating the association between the SRLV sequences analysis and histopathological lesions observed in different tissues of clinically affected goats from Poland. No significant correlation was observed between histological features and alterations in the sequences from different tissues. No tissue-specific signature pattern was identified. It was shown that the provirus load is positively correlated with lesion severity in various SRLV-affected tissues. The results of the phylogenetic analysis revealed the existence of a putative new subtype which should lead to the consideration of an update of the current SRLV classification. The A27 subtype in goats induces histopathological changes that are typical of SRLVs in general and contribute toward ongoing interspecies transmission and viral persistence. Our results indicated that the relationship between small ruminants and SRLVs is complex and that the severity of infection varies depending on the viral strain and the genetic background of the host. More extensive studies, including the isolation and characterization of the biological properties of these viruses, should be performed to evaluate their pathogenic potential.

The findings of this study have to be seen in light of some limitations. The number of tested animals was small (n = 6); all goats originated from the same flock and all animals were infected with the same SRLV subtype (A27). Furthermore, a limitation of this study is undoubtedly a lack of all *env* gene sequences.

## Figures and Tables

**Figure 1 viruses-14-00735-f001:**
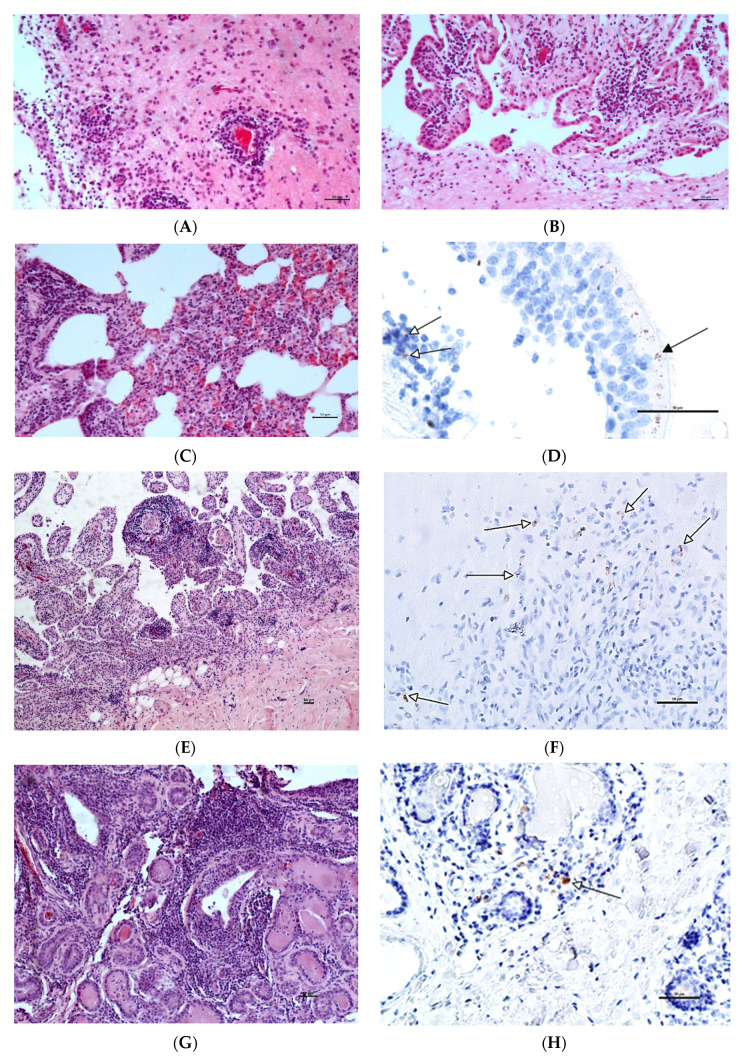
Example photomicrographs of tissue sections from the CAEV-infected goats. (**A**–**C**,**E**,**G**)—Sections stained with hematoxylin and eosin (HE); (**D**,**F**,**H**)—Immunohistochemical (IHC) labeling for CAEV p28 detection. (**A**) Brainstem, goat 5. Locally present mononuclear infiltrations around the blood vessel; (**B**) Choroid plexus, goat 5. Mononuclear infiltration within the choroid; (**C**) Lung, goat 7. Moderate interstitial pneumonia. The alveolar septa and peribronchial interstitium are widened by diffuse mononuclear infiltration and eosinophilic amorphic material; (**D**) Lung, goat 7. Positive CAEV p28 immunolabeling visible as brown particles in the cytoplasm of the bronchiolar epithelium (black arrows) and single macrophage-like cells (white arrows); (**E**) Carpal joint, goat 5. Hypertrophic synovial membrane-forming villous projections thickened by mononuclear infiltrates; (**F**) Carpal joint, goat 7. Positive CAEV p28 immunolabeling visible as brown particles in the cytoplasm of singular cells (arrows) within subsynovial mononuclear infiltrations; (**G**) Mammary gland, goat 5. Multifocal to diffuse mononuclear infiltrations around lactiferous ducts, blood vessels and within interstitial tissue; (**H**) Mammary gland, goat 5. Positive CAEV p28 immunolabeling in the cytoplasm of macrophage-like cells infiltrating the periacinar area; Scale bars = 50 µm.

**Figure 2 viruses-14-00735-f002:**
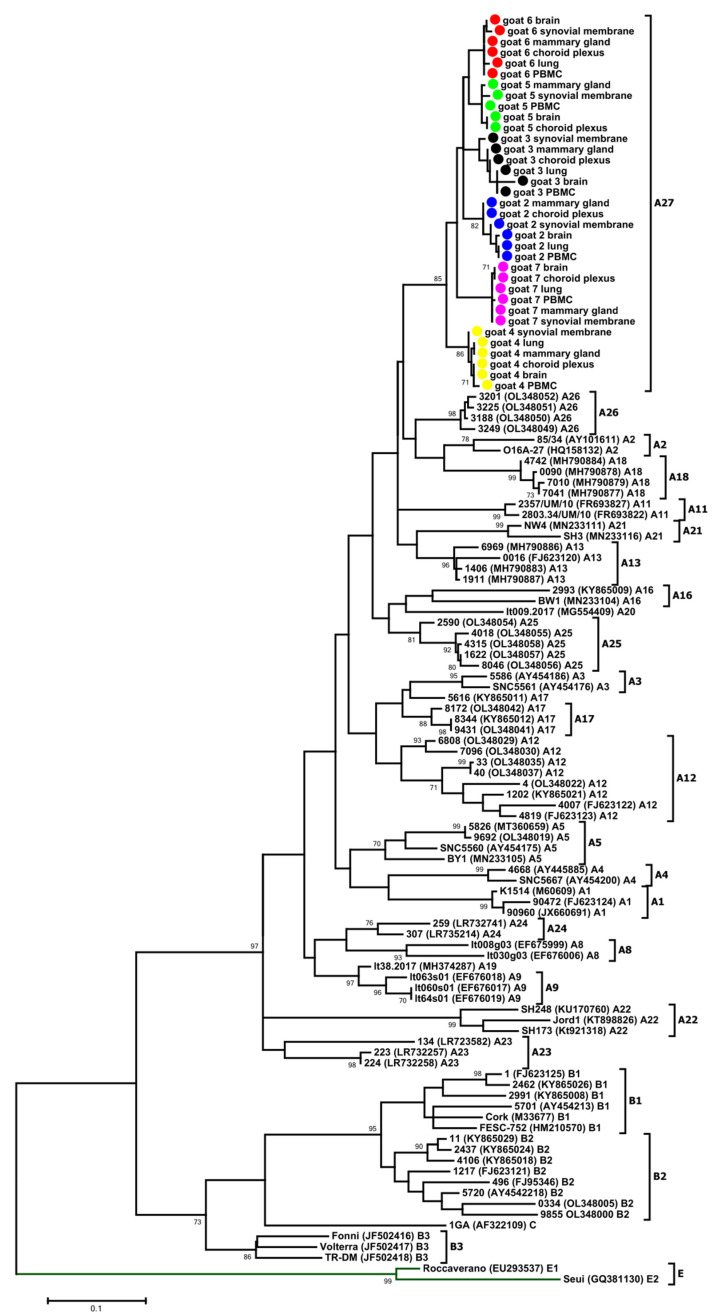
Maximum-likelihood phylogenetic tree based on the alignment of the CA fragment of the *gag* gene. Sequences from this study are labeled by black circles. Numbers at the branches indicate the percentage of bootstrap values obtained from 1000 replicates. Bootstrap values > 70% are shown. In the present study, the SRLV subtypes found by Colitti et al. [23] were renamed from A18 to A19 and from A19 to A20, and the subtypes found by Olech et al. [8] were renamed from A23 to A25 and from A24 to A26; goat 2—Blue, goat 3—Black, goat 4—Yellow, goat 5—Green, goat 6—Red, goat 7—Pink.

**Figure 3 viruses-14-00735-f003:**
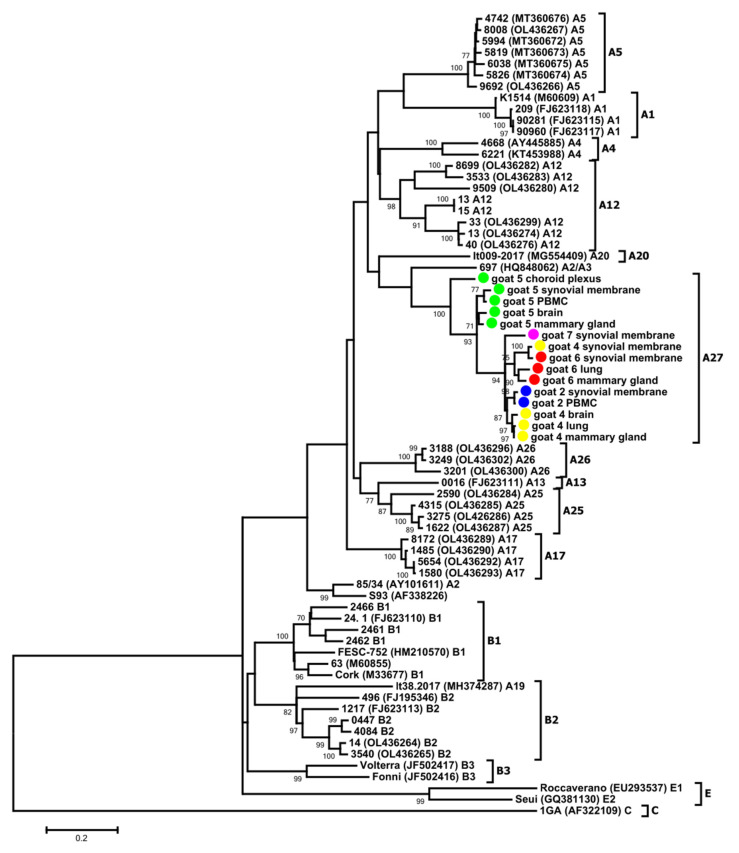
Maximum-likelihood phylogenetic tree based on the alignment of the V4V5 fragment of the *env* gene. Sequences from this study are labeled by black circles. Numbers at the branches indicate the percentage of bootstrap values obtained from 1000 replicates. Bootstrap values >70% are shown. In the present study, the SRLV subtypes found by Olech et al. [8] were renamed from A23 to A25 and from A24 to A26.

**Figure 4 viruses-14-00735-f004:**
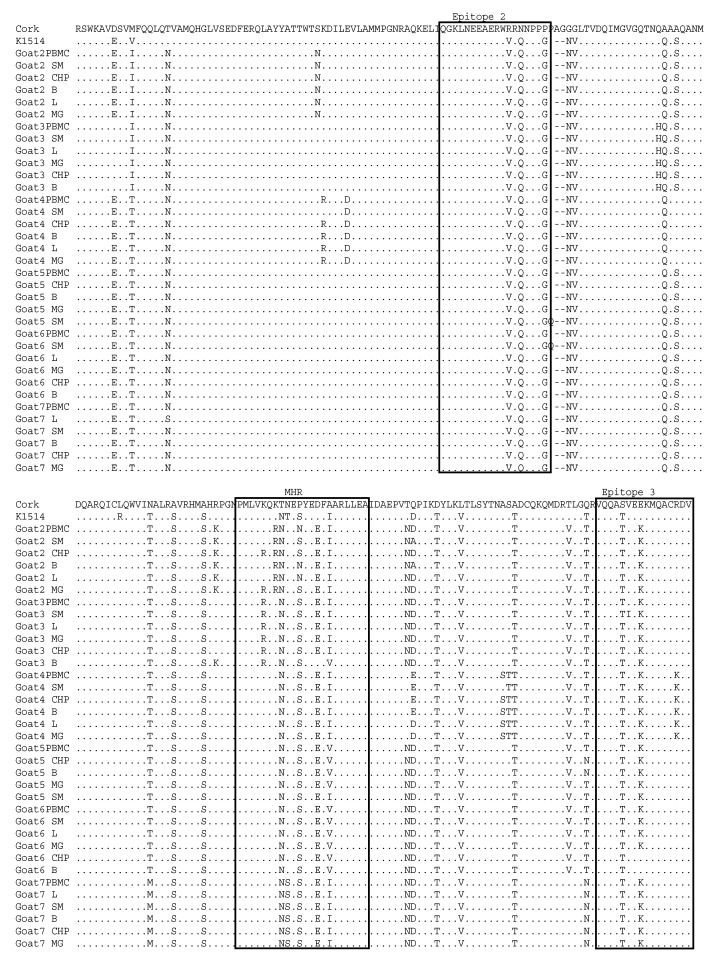
Alignment (MUSCLE) of deduced amino acid sequences of the *gag*-p25 capsid protein of SRLVs obtained in this study and K1514 (GenBank accession number M60609) and Cork (GenBank accession number M33677) reference strains, which are MVV (group A) and CAEV (group B) prototype strains, respectively. Identical residues are indicated by dots (.). Two immunodominant epitopes and the major homology region (MHR) are within squares. PBMC—Peripheral blood mononuclear cells, L—Lung, SM—Synovial membrane, B—Brain, CHP—Choroid plexus, MG—Mammary gland.

**Figure 5 viruses-14-00735-f005:**
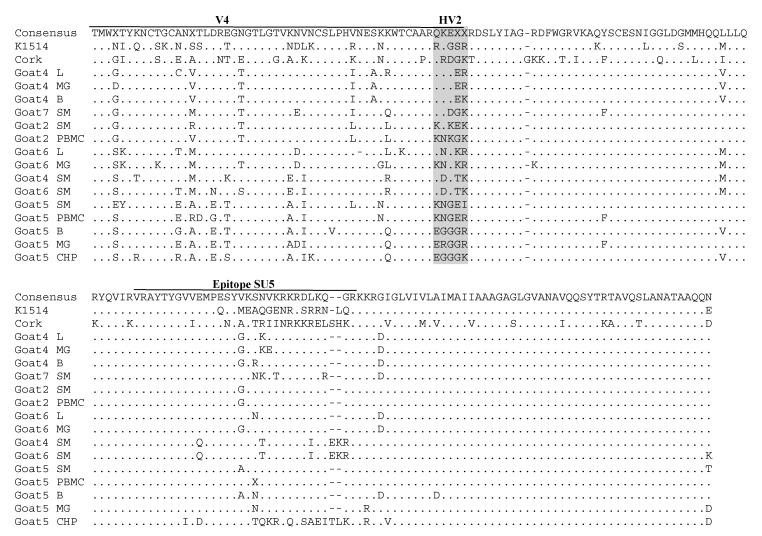
Alignment of deduced amino acid sequences of an immunodominant epitope of the ENV protein and variable region V4 of SRLVs obtained in this study and reference strains. Deletions are indicated by a dash (-), and identical residues are indicated by dots (.). PBMC—Peripheral blood mononuclear cells, L—Lung, SM—Synovial membrane, B—Brain, CHP—Choroid plexus, MG—Mammary gland.

**Figure 6 viruses-14-00735-f006:**
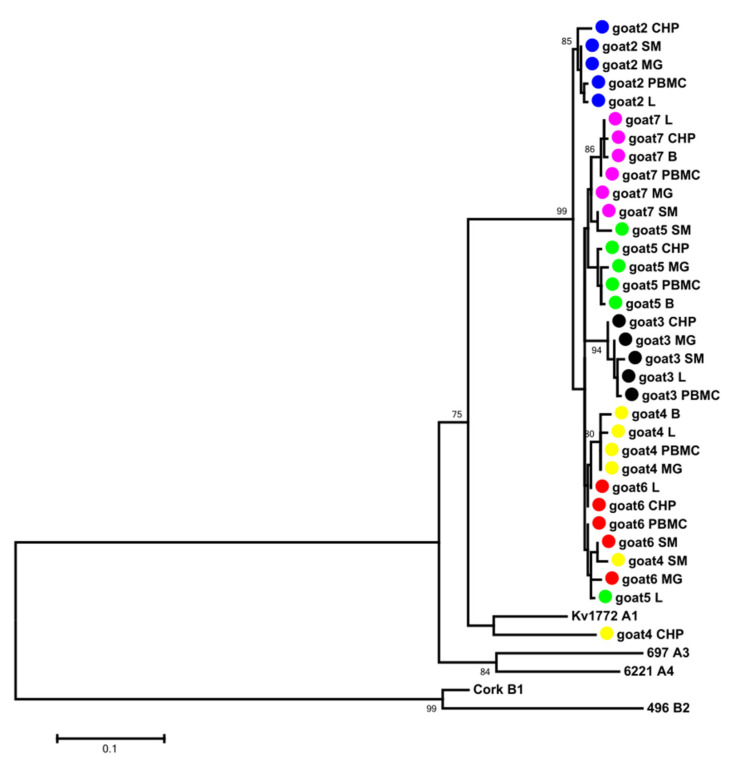
Maximum-likelihood phylogenetic tree based on the alignment of the LTR fragment. Sequences from this study are labeled by black circles. Numbers at the branches indicate the percentage of bootstrap values obtained from 1000 replicates. Bootstrap values >70% are shown. PBMC—Peripheral blood mononuclear cells, L—Lung, SM—Synovial membrane, B—Brain, CHP—Choroid plexus, MG—Mammary gland, goat 2—Blue, goat 3—Black, goat 4—Yellow, goat 5—Green, goat 6—Red, goat 7—Pink.

**Figure 7 viruses-14-00735-f007:**
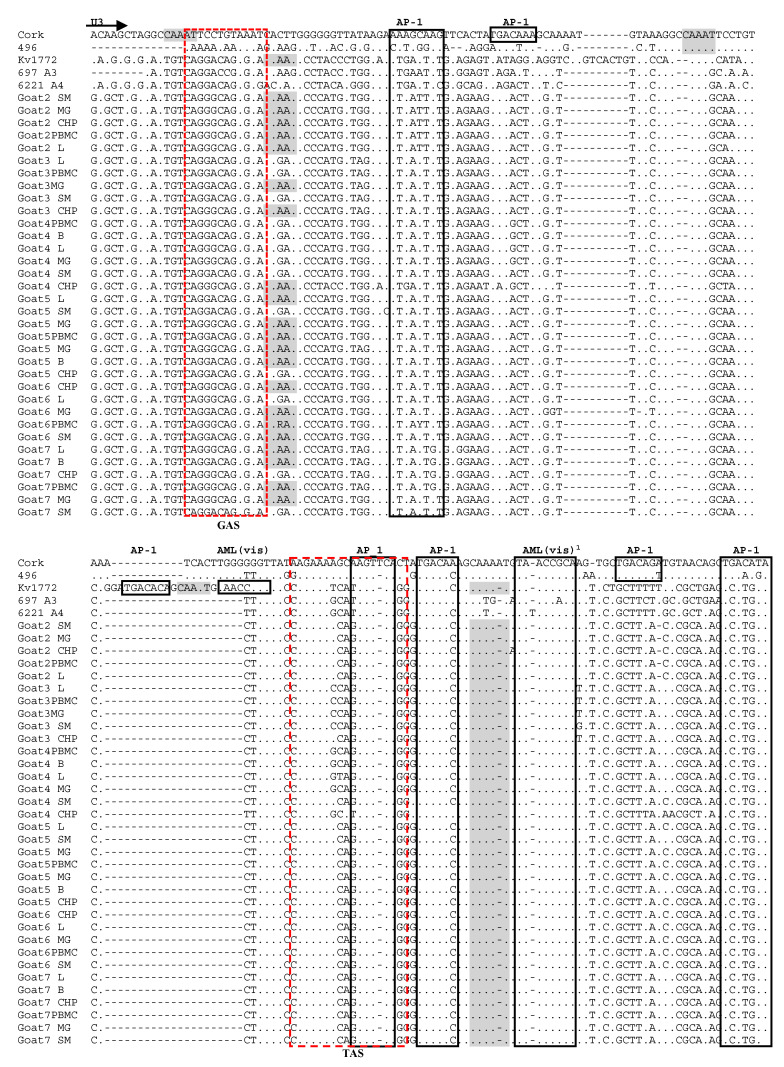
Alignment of U3-R sequences of LTR region from Polish SRLV strains. Sequences are aligned against prototype strains K1514 and Cork representative of SRLV groups A and B, respectively. Dots indicate identity with Cork, and dashes represent gaps. Boundaries between U3, R and U5 are indicated by straight arrows. AP-1, AP-4, AML (vis) motifs, TATA box, GAS, TAS and polyadenylation signal (poly A) are marked by boxes. Gray boxes represent CAAAT sequences. PBMC—Peripheral blood mononuclear cells, L—Lung, SM—Synovial membrane, B—Brain, CHP—Choroid plexus, MG—Mammary gland.

**Figure 8 viruses-14-00735-f008:**
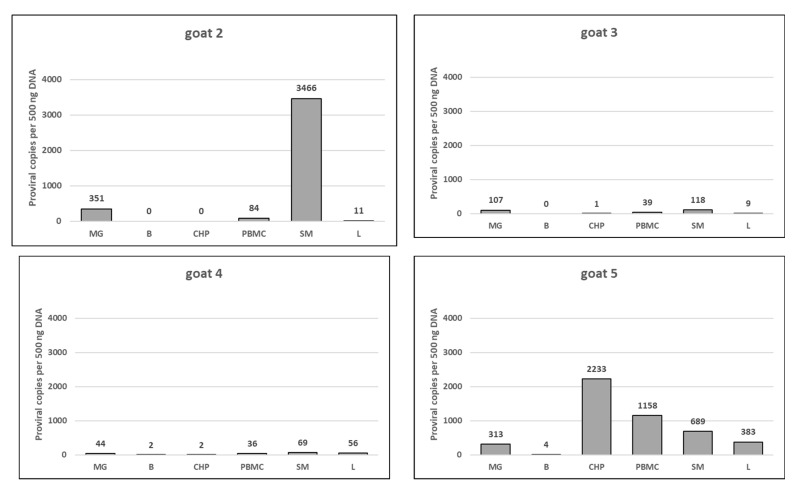
Quantification of SRLV proviral load in different organs and cells of goats analyzed in this study. PBMC—Peripheral blood mononuclear cells, L—Lung, SM—Synovial membrane, B—Brain, CHP—Choroid plexus, MG—Mammary gland. Mean value of SRLV proviral copy number is shown above the bar.

**Table 1 viruses-14-00735-t001:** Inflammation intensity scored in histopathological sections from the examined goats.

	Joints	Brains	Lung	Mammary Glands
Goat 2	3 *	0	1	2
Goat 3	2 *	0	1	2
Goat 4	2 *	0	1	3
Goat 5	3 *	2	1	2
Goat 6	2 *	0	1	2
Goat 7	3 *	2	2	2

* arthritis score according to the grading system by Cheevers et al. [21].

**Table 2 viruses-14-00735-t002:** Mean pairwise nucleotide genetic distance between subtypes of genotype A (inter-genotype) based on the CA fragment of the *gag* gene. In the present study, the SRLV subtypes found by Colitti et al. [23] were renamed from A18 to A19 and from A19 to A20, and the subtypes found by Olech et al. [8] were renamed from A23 to A25 and from A24 to A26.

	A1	A2	A3	A4	A5	A8	A9	A11	A12	A13	A16	A17	A18	A19	A20	A21	A22	A23	A24	A25	A26	A27
A1	-	-	-	-	-	-	-	-	-	-	-	-	-	-	-	-	-	-	-	-	-	-
A2	16.6	-	-	-	-	-	-	-	-	-	-	-	-	-	-	-	-	-	-	-	-	-
A3	13.8	11.5	-	-	-	-	-	-	-	-	-	-	-	-	-	-	-	-	-	-	-	-
A4	14.7	16.1	15.3	-	-	-	-	-	-	-	-	-	-	-	-	-	-	-	-	-	-	-
A5	15.7	14.7	11.9	15.8	-	-	-	-	-	-	-	-	-	-	-	-	-	-	-	-	-	-
A8	16.2	17.1	15.3	17.3	17.2	-	-	-	-	-	-	-	-	-	-	-	-	-	-	-	-	-
A9	15.2	14.5	12.2	16.5	14.3	15.1	-	-	-	-	-	-	-	-	-	-	-	-	-	-	-	-
A11	15.9	14.6	14.6	18.4	15.9	15.9	13.6	-	-	-	-	-	-	-	-	-	-	-	-	-	-	-
A12	16.2	12.7	10.9	16.6	13.5	16.7	15.8	14.5	-	-	-	-	-	-	-	-	-	-	-	-	-	-
A13	16.0	10.8	11.9	14.9	13.8	17.5	16.0	14.7	13.3	-	-	-	-	-	-	-	-	-	-	-	-	-
A16	17.9	15.9	17.2	15.7	16.4	19.6	20.6	18.0	16.1	16.3	-	-	-	-	-	-	-	-	-	-	-	-
A17	14.2	11.5	8.8	15.0	11.0	15.7	11.5	15.5	10.0	13.2	14.7	-	-	-	-	-	-	-	-	-	-	-
A18	19.3	11.9	14.2	18.3	16.2	20.4	17.5	17.1	14.4	12.3	15.9	13.6	-	-	-	-	-	-	-	-	-	-
A19	14.8	13.2	10.9	14.5	12.3	14.3	4.6	12.8	14.8	14.1	18.1	10.5	15.9	-	-	-	-	-	-	-	-	-
A20	19.3	13.7	13.5	15.7	13.5	16.9	16.7	16.0	13.3	14.1	15.2	14.5	14.9	14.8	-	-	-	-	-	-	-	-
A21	15.8	13.8	12.6	18.1	15.4	18.0	16.1	13.5	12.9	12.5	16.9	13.7	14.5	15.0	15.9	-	-	-	-	-	-	-
A22	24.4	20.6	19.7	24.2	18.6	24.0	22.5	22.7	20.5	22.6	23.3	20.2	21.0	22.2	18.8	21.3	-	-	-	-	-	-
A23	19.6	17.2	16.4	18.8	17.4	17.0	15.7	14.9	16.9	18.8	21.7	16.8	20.3	14.5	15.9	19.4	21.3	-	-	-	-	-
A24	15.9	12.6	12.0	16.0	14.2	13.2	11.8	13.9	14.0	13.3	18.2	12.8	15.6	12.5	16.2	13.1	21.1	15.0	-	-	-	-
A25	15.1	11.1	12.3	15.6	14.0	16.2	14.0	15.3	12.7	11.9	15.8	12.6	12.2	13.3	11.6	14.6	21.0	17.5	14.4	-	-	-
A26	17.7	11.0	12.9	16.6	13.2	17.2	15.5	14.4	13.0	11.1	16.0	13.0	10.2	14.8	11.4	13.1	18.8	17.6	12.1	10.3	-	-
A27	13.2	11.3	11.9	16.0	13.0	17.4	14.7	13.2	12.4	10.0	14.2	12.4	12.8	13.6	14.0	12.2	21.7	18.2	14.2	11.0	10.7	-

## Data Availability

All data generated and analyzed in this study are included in this article.

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
