# Peer review of "Molecular Characterization of Small Ruminant Lentiviruses Isolated from Polish Goats with Arthritis"

_viruses, 2022, doi:10.3390/v14040735_

Round 1

Reviewer 1 Report

General considerations:

The submitted manuscript is very well written and presents a detailed analysis of six naturally SRLV infected goats showing the typical symptoms of SRLV-induced clinical arthritis. This is the first study of SRLV infected, arthritic goats in Poland and contributes to our understanding of the virulence of SRLV-A in goats. The authors performed excellent work dissecting the phylogenetical characteristics of the infecting viruses, analyzing the crucial targets of humoral immunity in their genes, and dissecting the potential differences between the transcription factors binding sites in the LTR of these viruses and those of previous characterized SRLV. Finally, they completed their work with histopathological analysis of different tissues, the immunohistochemical demonstration of Gag antigen in these tissues, and quantitative analysis of the proviral load in the selected sites. The results are high quality, and their interpretation and discussion are sound. The abstract perfectly summarize the results obtained and their careful interpretation. A few points may be discussed, and suggestions can be found in the specific criticism session.

Specific criticism:

Title: “Molecular and Pathological Characterization of Small Ruminant Lentiviruses Isolated from Polish Goats with Arthritis”

I don’t think it is appropriate to speak about a “pathological” characterization of a virus. Pathological may be deleted.

Lines 51 – 54: “On infection, animals may remain asymptomatic carriers for life, and only a small proportion of infected animals develop clinical signs such interstitial pneumonia, indurative mastitis (“hard udder”), arthritis, dyspnea, and more rarely encephalitis, ataxia or paralysis [1].”

It may be appropriate to adapt the sequence of symptoms to the studied species, i.e., the goat: arthritis, mastitis, rarely and only in young kits encephalitis and, mainly in a subclinical form, interstitial pneumonia with dyspnea.  

Lines 65 – 67: “In general, isolates from group A (MVV-like) lead to mastitis and pneumonia while arthritis and nervous signs are more characteristic for strains belonging to group B (CAEV-like).”

This sentence may be modified according to the literature. In fact, it appears that the jury is still out on this point. The paper by Glaria et al. (reference 25) suggests that the genotype of the virus, in this case, SRLV-B2, determines the clinical outcome, i.e., arthritis. In reference 32 (Grego et al.), however, compelling evidence is presented, indicating that it is not the genotype of the virus but rather the genotype of the host or even the infected species, in this case, sheep, that determines the clinical outcome. The seven isolates It-128, It-169, It-170, It-172, and It-2038, It-585, and It-561, clustered within the SRLV-B (CAEV) genotype; this notwithstanding, the animals showed typical gross and histopathological lesions compatible with an SRLV-A (Maedi Visna Virus) infection. The only exception was the Pi1 isolate inducing arthritis in the infected sheep.

Line 80 and discussion, lines 413 – 463: a detailed study of the LTR of an SRLV attenuated strain points to the difficulty in relating the importance of transcription binding sites to the virulence and tissue tropism of these viruses (Blatti-Cardinaux L. et al. Journal of General Virology (2016), 97, 1699–1708. The results obtained and thoroughly discussed by the authors appear to support this conclusion.

Line 124: Looking at the p28 expressing cells, it may be possible to name these cells as “macrophage-like” rather than “inflammatory cells”, as done in the figure legend.

Lines 167 – 174 and Table 2: The subdivision of these sequences in new clusters or subgroups should be tested according to the rules used for HIV and introduced by Shah et al. (Virology 319 (2004) 12–26) in the SRLV field. The proposed cutoff to define a new subgroup is 15%, and I am not sure that the new A27 subgroup is sufficiently distant from other subtypes, such as A13 or A25, to qualify as a new subgroup. This depends on the gene selected for the analysis. Indeed, the distances observed in the highly variable region of the env gene (lines 198 – 200) appear to confirm the classification proposed by the authors.  

Lines 393 – 394: This sentence is unclear, and I don’t understand the use of “respectively” in this context. In contrast, I agree entirely with the conclusions of the authors in lines 396 – 398.

Line 488: reference #6 may be added to this list.

Discussion: the discussion is well written but rather lengthy and may be edited to improve its impact.

Author Response

We would like thank the reviewer for his comments on our manuscript. We have acted upon the suggestions provided by the reviewer  and alterations were included in the updated version of the manuscript.

Reviewer 1

General considerations:

The submitted manuscript is very well written and presents a detailed analysis of six naturally SRLV infected goats showing the typical symptoms of SRLV-induced clinical arthritis. This is the first study of SRLV infected, arthritic goats in Poland and contributes to our understanding of the virulence of SRLV-A in goats. The authors performed excellent work dissecting the phylogenetical characteristics of the infecting viruses, analyzing the crucial targets of humoral immunity in their genes, and dissecting the potential differences between the transcription factors binding sites in the LTR of these viruses and those of previous characterized SRLV. Finally, they completed their work with histopathological analysis of different tissues, the immunohistochemical demonstration of Gag antigen in these tissues, and quantitative analysis of the proviral load in the selected sites. The results are high quality, and their interpretation and discussion are sound. The abstract perfectly summarize the results obtained and their careful interpretation. A few points may be discussed, and suggestions can be found in the specific criticism session.

Specific criticism:

Title: “Molecular and Pathological Characterization of Small Ruminant Lentiviruses Isolated from Polish Goats with Arthritis”

I don’t think it is appropriate to speak about a “pathological” characterization of a virus. Pathological may be deleted.

Re: It has been corrected.

Lines 51 – 54: “On infection, animals may remain asymptomatic carriers for life, and only a small proportion of infected animals develop clinical signs such interstitial pneumonia, indurative mastitis (“hard udder”), arthritis, dyspnea, and more rarely encephalitis, ataxia or paralysis [1].”

It may be appropriate to adapt the sequence of symptoms to the studied species, i.e., the goat: arthritis, mastitis, rarely and only in young kits encephalitis and, mainly in a subclinical form, interstitial pneumonia with dyspnea.  

Re: It has been corrected.

Lines 65 – 67: “In general, isolates from group A (MVV-like) lead to mastitis and pneumonia while arthritis and nervous signs are more characteristic for strains belonging to group B (CAEV-like).”

This sentence may be modified according to the literature. In fact, it appears that the jury is still out on this point. The paper by Glaria et al. (reference 25) suggests that the genotype of the virus, in this case, SRLV-B2, determines the clinical outcome, i.e., arthritis. In reference 32 (Grego et al.), however, compelling evidence is presented, indicating that it is not the genotype of the virus but rather the genotype of the host or even the infected species, in this case, sheep, that determines the clinical outcome. The seven isolates It-128, It-169, It-170, It-172, and It-2038, It-585, and It-561, clustered within the SRLV-B (CAEV) genotype; this notwithstanding, the animals showed typical gross and histopathological lesions compatible with an SRLV-A (Maedi Visna Virus) infection. The only exception was the Pi1 isolate inducing arthritis in the infected sheep.

Re: We agree with the reviewer. The authors decided to remove the sentence.

Line 80 and discussion, lines 413 – 463: a detailed study of the LTR of an SRLV attenuated strain points to the difficulty in relating the importance of transcription binding sites to the virulence and tissue tropism of these viruses (Blatti-Cardinaux L. et al. Journal of General Virology (2016), 97, 1699–1708. The results obtained and thoroughly discussed by the authors appear to support this conclusion.

Re: Our results are in line with results obtained by Blatti-Cardinaux et al. The relevant citation has been included.

Line 124: Looking at the p28 expressing cells, it may be possible to name these cells as “macrophage-like” rather than “inflammatory cells”, as done in the figure legend.

Re: It has been corrected.

Lines 167 – 174 and Table 2: The subdivision of these sequences in new clusters or subgroups should be tested according to the rules used for HIV and introduced by Shah et al. (Virology 319 (2004) 12–26) in the SRLV field. The proposed cutoff to define a new subgroup is 15%, and I am not sure that the new A27 subgroup is sufficiently distant from other subtypes, such as A13 or A25, to qualify as a new subgroup. This depends on the gene selected for the analysis. Indeed, the distances observed in the highly variable region of the env gene (lines 198 – 200) appear to confirm the classification proposed by the authors.  

Re: Shah et al. performed phylogenetic analysis based on 1.8 kb fragment of the gag-pol region and 1.2 kb fragment in the pol region. In our paper we do not based on criteria (cut off 15%) proposed by Shah et al. because in our paper classification of SRLVs was performed on a ~0.4 kb gag fragment which is shorter and more conservative than fragments used by Shah et al. Looking at the table you can see that many of the subtypes do not meet the criteria proposed by Shah et al. The affiliation of the new cluster, A27, was supported with high bootstrap values and phylogenetic analysis on the basis of env gene confirmed that the sequences originating from the goats analyzed in this study belonged to the new subtype A27.

Lines 393 – 394: This sentence is unclear, and I don’t understand the use of “respectively” in this context. In contrast, I agree entirely with the conclusions of the authors in lines 396 – 398.

Re: It has been corrected.

Line 488: reference #6 may be added to this list.

Re: It has been added.

Discussion: the discussion is well written but rather lengthy and may be edited to improve its impact.

Re: Because the reviewer did not indicate what should be shortened in the discussion and because other reviewers had no comments on the discussion the authors decided not to revise it.

Reviewer 2 Report

Dear Editor!

The paper submitted for review under the title “Molecular and Pathological Characterization of Small Ruminant Lentiviruses Isolated from Polish Goats with Arthritis” by authors: Olech M. et al. submitted to section: Animal Viruses, in Viruses, is a diligently conducted new study on the mechanism of structure elucidation of the molecular properties of SRLVs isolated from different organs of six arthritic goats from Poland.

The manuscript is clearly written and technically sound.

The methods are appropriate and adequately conducted.

The manuscript may be supplemented with text clarifying the limitation of this study.

Technical errors in the text: Page 24, line 766-767.

Author Response

We would like thank the reviewer for his comments on our manuscript. We have acted upon the suggestions provided by the reviewer  and alterations were included in the updated version of the manuscript.

Reviewer 2

The paper submitted for review under the title “Molecular and Pathological Characterization of Small Ruminant Lentiviruses Isolated from Polish Goats with Arthritis” by authors: Olech M. et al. submitted to section: Animal Viruses, in Viruses, is a diligently conducted new study on the mechanism of structure elucidation of the molecular properties of SRLVs isolated from different organs of six arthritic goats from Poland.

The manuscript is clearly written and technically sound.

The methods are appropriate and adequately conducted.

The manuscript may be supplemented with text clarifying the limitation of this study.

Re: The limitation of the study has been included.

Technical errors in the text: Page 24, line 766-767.

Re: It has been corrected.